# Direct or Indirect ESPT Mechanism in CFP psamFP488? A Theoretical-Computational Investigation

**DOI:** 10.3390/ijms232415640

**Published:** 2022-12-09

**Authors:** Greta Donati, Nadia Rega

**Affiliations:** 1Department of Chemical Sciences, University of Napoli Federico II, Complesso Universitario di M.S. Angelo, Via Cintia 21, I-80126 Napoli, Italy; 2Scuola Superiore Meridionale, Largo San Marcellino 10, I-80138 Napoli, Italy

**Keywords:** fluorescent proteins, photo-induced proton transfer, excited state ab initio molecular dynamics

## Abstract

Fluorescent Proteins are widely studied for their multiple applications in technological and biotechnological fields. Despite this, they continue to represent a challenge in terms of a complete understanding of all the non-equilibrium photo-induced processes that rule their properties. In this context, a theoretical-computational approach can support experimental results in unveiling and understanding the processes taking place after electronic excitation. A non-standard cyan fluorescent protein, psamFP488, is characterized by an absorption maximum that is blue-shifted in comparison to other cyan fluorescent proteins. This protein is characterized by an extended Stokes shift and an ultrafast (170 fs) excited state proton transfer. In this work, a theoretical-computational study, including excited state ab initio dynamics, is performed to help understanding the reaction mechanism and propose new hypotheses on the role of the residues surrounding the chromophore. Our results suggest that the proton transfer could be indirect toward the acceptor (Glu167) and involves other residues surrounding the chromophore, despite the ultrafast kinetics.

## 1. Introduction

The complex world of fluorescent proteins (FP) is an intriguing and fascinating field of chemical research.

FPs are small and intrinsically fluorescent systems in which the chromophore forms autocatalytically in a host-independent process. These are among the unique features of FPs that permit their employment in a wide range of applications in cellular and molecular biology and biotechnology, along with the fact that their properties can be tuned [1].

The growing interest in this class of proteins was first triggered by the discovery of the Green Fluorescent Protein in the jellyfish *Aequorea Victoria*, which initiated the so-called GFP revolution [2]. An eleven-stranded β-barrel structure and chromogenic XYG peptide are common in GFP-like proteins [3]. Various color emitters are obtained by extending the chromophore conjugation length or tuning the 3D solvation shell, affecting both the cis–trans isomerization and the chromophore protonation state [3,4]. Further developments through mutagenesis can enhance the stability and expand the available spectral range, allowing the potential of GFP and related fluorescent proteins to be exploited [5]. Despite a huge number of applications, FPs are complex systems, and their photochemistry is not yet completely understood.

Nowadays, thanks to modern spectroscopic techniques such as picosecond transient Raman Spectroscopy, time-resolved fluorescence [6], femtosecond-stimulated Raman Spectroscopy [7], and femtosecond UV spectroscopy [8], it is possible to obtain major insights into photo-induced processes. The most studied system in the field of photo-reactivity is GFP, on account of the great interest in its huge number of applications and the complexity of its photochemistry. The chromophore *p*-hydroxybenzylideneimidazolinone in its neutral form (HBDI; see Figure 1) is responsible for an absorption band around 398 nm.

Upon electronic excitation, an excited-state proton transfer (ESPT) reaction takes place involving a hydrogen bond network composed of the chromophore, a crystallographic water molecule, and the residues Ser205 and Glu222 [9]. It is the anionic form of the chromophore obtained from this reaction that is responsible for the bright fluorescence around 508 nm [10]. Although many experimental and theoretical studies have been performed to clarify driving force of the ESPT along with its mechanism and kinetics, many open hypotheses exist and the available experimental data are insufficient to provide a complete picture of these phenomena. In this context, the theoretical-computational approach can be crucial to help in understanding the driving forces induced by the electronic excitation favoring the ESPT [11,12,13,14]. Despite this, thanks to their huge number of applications, studies on FPs are growing day by day, representing a great challenge from a scientific point of view.

The discovery in *Anthozoa* of an entire family of fluorescent proteins and non-fluorescent chromoproteins distantly related to the GFP from *Aequorea Victoria* has provided new tools that can be alternative or complement to the existing uses of GFP [15]. Fluorescent proteins isolated from coral reef organisms in the *Anthozoa* class can be grouped into five classes: red, yellow, cyan, green, and non-fluorescent chromoproteins. The general conclusion of the several studies on a variety of coral FPs is that the most important determinant of emission color is the physical extent of the conjugated portion of the chromophore. However, local environmental effects such as the position of the charged group near the chromophore, can lead to remarkable shifts in absorbance and emission maxima [16,17,18]. Because of the high heterogeneity of the colors of such coral FPs, these systems are very interesting for a variety of in vivo techniques; moreover, they can be employed as models for experimental studies on the evolution of protein families [15].

Within this family, the class of cyan fluorescent proteins (CFPs) possesses the same chromophore as the green fluorescent proteins [16]. These proteins typically have an emission peak between 485–495 nm, although more blue-shifted variants can occasionally be found down to 477 nm, and exhibit absorption maxima from 430 and 460 nm. A cyan fluorescent protein exhibiting different behavior in comparison to other cyan fluorescent proteins, termed psamFP488, has been isolated from the genus *Psammocora* of reef building corals [19]. In particular, in this protein, unlike other cyan fluorescent proteins, a very blue-shifted absorption maximum is displayed around 404 nm, while the emission range is one typical of CFPs. Therefore, psamFP488 is characterized by an extended Stokes shift that is unexpected for CFPs. Moreover, the absorption maximum is very similar to that observed for wtGFP, making it reasonable to hypothesize similarities with the photo-chemical behavior of GFP.

In their work [19], Kennis et al. performed experiments based on transient absorption spectroscopy and pump-dump-probe spectroscopy, and suggested that an ESPT reaction takes place upon electronic excitation that converts the neutral chromophore in its anionic form to the species responsible for the observed fluorescence peak.

The observed large Stokes shift should find its origin in this ESPT process. Furthermore, time-resolved experiments have revealed an ultrafast kinetics correlated to both the ESPT and the ground state proton back-transfer. In particular, experimental findings suggest that the ESPT reaction occurs in 170 fs, followed by a proton back-transfer that closes the photocycle and returns to the neutral chromophore in 110 fs.

A scheme of the hypothesized photocycle for psamFP488 is shown in Figure 2.

Although an ESPT has been hypothesized, the mechanism is not clear. It is possible that the ultrafast proton shuttling in the excited and ground states results from a favorable hydrogen-bonding geometry between the donor and acceptor. The hypothesis of a network surrounding the chromophore was proposed by Kennis et al. on the basis of the homologous zFP538-K66M shown in Figure 3 because of the lack of a crystallographic structure in psamFP488.

The main difference between psamFP488 and zFP538-K66M is represented by residue 167; in psamFP488, the methionine 167 is replaced by a glutammic acid. It is reasonable to suppose that Glu167, which seems to be very close to the chromophore, can act as proton acceptor. Previously, it has been noted that position 167 is crucial to tuning the cyan emission [20].

However, the absence of a crystallographic structure in psamFP488, the intrinsic complexity in describing an excited state reaction, and the paucity of literature (both experimental and theoretical) on this system combine to make the current hypotheses and understanding of the reaction mechanism an open issue. An important contribution to provide insight on these problems, in particular with respect to obtaining a molecular picture of the experimental results, can be given by a theoretical-computational study.

In this study, a mixed quantum mechanics–molecular mechanics (QM/MM) approach is used to simulate the ESPT reaction by ab initio molecular dynamics in order to help in understanding the reaction mechanism starting from the hypotheses based on the experimental results.

In particular, because of the absence of a crystallographic structure in psamFP488, it is challenging to establish a reasonable starting configuration that can represent the first reactant of the photocycle; here, a computational approach can be helpful in hypothesizing plausible structural arrangements for the active site. In this way, different possibilities can be proposed and compared in order to obtain a reasonable starting point for the photocycle study.

Recently, it has been demonstrated at the theory level that a density functional theory (DFT) in its time-independent and time-dependent (TD) versions is able to provide a good description of theESPT reaction mechanism of GFP and the optical behavior of the chromophore in several environments [12]. Encouraged by these results, we employ a similar protocol here to describe a novel and as yet not well understood system in psamFP488. In this case, the challenge is to characterize the photoactive species involved in the reaction starting from a reasonable hypothesis of the protein active site while building a solid model of the reactant despite the absence of a crystallographic protein structure. Our results suggest that, although an ultrafast ESPT could favor the hypothesis of a direct proton transfer from the chromophore to the final acceptor, other possible scenarios should be considered as well. Indeed, we suggest that a more complex and indirect mechanism could be both plausible and compatible with this CFP photo-induced reaction.

The rest of this paper is organized as follows: the main results are presented in the Results and Discussion section; the theoretical-computational strategy is described in the Materials and Methods section; finally, our main conclusions are provided in the Conclusions section.

## 2. Results and Discussion

### 2.1. Characterization of the Cyan Fluorescent Protein Photocycle

#### 2.1.1. The Chromophore in Its Neutral Form; Part I: The Experimental Model

In this section, we discuss the characterization of the CFP chromophore in its neutral form, which is responsible for the ESPT reaction upon electronic excitation. By a chromophore in its neutral form we mean one that is characterized by a protonated phenolic ring. This first step of characterization covers an important part of our work for many reasons. First, recall that as we lack a reference conformation of the residues involved in the reaction due to the absence of the crystallographic structure of the protein, we have no representative initial configuration of the network. Second, the ESPT in FPs is not provided merely by the intrinsic photoacidity of the chromophore; in GFP the binding pocket has a crucial role in triggering the proton shuttle influencing its kinetics and mechanism as well [11,13,21]. For these reasons, guided by the experimental hypotheses, it is necessary to evaluate different possible network arrangements around the neutral psam FP488 chromophore. Therefore, it is quite a challenge to find an adequate species that rightly describes the protein active site. The complexity of this problem is further increased by the size of the system (with a huge number of degrees of freedom) and the intrinsically peculiar nature of the photocycle.

In order to find and characterize the reactant species, the closest residues involved in the reaction, and those involved in hydrogen bonding (HBs) with them, we started with a model proposed on the basis of experimental spectroscopic results [19]. This hypothesis was based on the possibility of direct ESPT between the chromophore and Glu167 reasonably representing the PT acceptor. Direct ESPT is corroborated by the ultrafast kinetics observed for this process (170 fs). In GFP, the chromophore gains a conformational freedom upon excitation that allows for significant HB network rearrangement, which in turn becomes more favorable for the reaction. This complex structural rearrangement is driven by a low-frequency mode (about 120 cm${}^{-1}$) that has previously been experimentally and theoretically analyzed [11,13,21]. As matter of fact, when considering the much faster ESPT kinetics in psamFP488 compared to GFP, it is reasonable to expect that in psamFP488 this complex rearrangement of the binding pocket is not necessary for the reaction, strongly suggesting a binding pocket arrangement predisposed to favor the reaction in the ground state. According to the structural arrangement, this mechanism presumes that the chromophore is hydrogen-bonded to both Glu167 and Ser148, as shown in Figure 4.

In the partition treated at the QM level of theory, we included a crystallographic water molecule, which is plausibly involved in a hydrogen bond with the chromophore. The ultrafast kinetics experimentally recorded for the ESPT suggests that the photoinduced reaction does not involve important structural rearrangements after the electronic excitation, which usually require from a few hundred femtoseconds to tens of picoseconds to take place. Therefore, we arranged the Glu167 conformation, in particular the carboxylate group orientation, in order to engage a hydrogen bond with the chromophore. Accordingly, the initial guess of the reactant structure was set up in order to allow hydrogen bonds between the chromophore, the Ser148, and the crystallographic water. This choice should favor prompt stabilization of the anionic form of the chromophore after the proton transfer.

Around the model shown in Figure 4, the remaining part of the protein was arranged as the crystallographic structure and treated at the lower level of theory. Starting from this guess, we performed an optimization procedure to obtain a minimum energy structure in the ground state to represent the neutral form, i.e., the reactant of the proton transfer reaction (form A in the photocycle shown in Figure 2). However, the structure quickly evolved to an anionic form during the optimization procedure. In other words, a proton was promptly transferred from the chromophore to the Glu167 during minimization. The obtained structure is shown in Figure 5.

This result suggests that this configuration is probably poorly representative of the stable neutral chromophore form in the ground electronic state and its hydrogen bond network. Table 1 shows the main structural parameters extracted from the optimized anionic structure. In particular, the intermolecular oxygen distances of the residues involved in the hydrogen bond network, inter- and intramolecular O-H distances, chromophore C-O distance, and hydrogen bond angles are presented.

From inspection of the results reported in Table 1, it is clear that Glu167 is protonated; indeed, a typical bond O-H distance (0.99 Å) is found for this residue. The intermolecular oxygen–oxygen distances show a strong hydrogen bond interaction between the chromophore and Glu167 O1 oxygen (see labels in Figure 5), while O2 remains more distant from it.

A hydrogen bond between Ser148 and the water molecule with the phenolate ring is retained, as can be seen from the intermolecular O-O and O-H distances. Here, we report several values of the HB${}_{acceptor}$-HB${}_{donor}$-H angle relative to the residues involved in hydrogen bonds, which are representative of the HB strength. The observed values of these angles confirm a strong interaction between the chromophore and the Ser148, water molecule, and Glu167, while the C-O distance of the phenolic ring is compatible with an anionic species.

The optimization results suggest that this protein environment strongly favors the anionic form, which is highly stabilized by the residues surrounding the chromophore through hydrogen bond interactions. In particular, it seems that the HB interactions with Ser148 and the water molecule strongly promote proton transfer, favoring a stable anionic form. On the other hand, Glu167 initially being in its unprotonated form and close to the phenolic ring makes for an optimal proton acceptor. Therefore, all the structural arrangements in this network hypothesis seem to strongly favor the anionic form.

On the basis of these considerations, we investigated other possible network arrangements while maintaining the assumption of a direct interaction between the chromophore and Glu167, which has been reasonably hypothesized by experimentalists on the basis of the observed ultrafast ESPT kinetics. This latter hypothesis indicates that a network favoring proton transfer must be already present and stable in the ground electronic state. For this reason, we tried to obtain a protonated chromophore form starting from a different network configuration; here, the key difference from the previous form is a different solvation arrangement around the chromophore tyrosine. This new hypothesis is presented in Figure 6.

Here, the main difference from the previous network involves the Ser148 arrangement, which shows the O-H group pointing at Glu167 and not at phenolic oxygen. This choice was made with the aim of reducing the chromophore solvation, and as a consequence reducing the stabilization of its anionic form. By analogy to other FPs, such as GFP, in which a proton is shuttled through a reaction path involving a Ser residue, this new network suggests a more important role of Ser148 in the ESPT reaction. Therefore, we reoriented Ser148 and slightly changed the Glu167 conformation to weaken its hydrogen bond interaction with the chromophore. This was done by acting on the Glu167 side-chain dihedral angles, taking care to adopt values in agreement with those typical of this residue.

This initial guess was used for a second optimization procedure, leading to the final structure shown in Figure 7; in this case, an anionic final form of the chromophore was obtained.

In Table 2, we list the main structural parameters extracted from the optimized structure.

The intermolecular O-O and O-H distances and the H(ser)-O(ser)-O(glu) angle between Ser148 and Glu167 suggest that a hydrogen bond interaction is formed during optimization. A shortening of the intermolecular oxygen distances (2.47 Å) is observed for the tyrosine and the O1 oxygen of Glu167, suggesting that a strong interaction with Glu167 is favored in this model, leading to proton transfer and a final anionic chromophore form. As in the previous case, we found a C-O phenolic ring distance of 1.28 Å, which is compatible with a an anionic form. This suggests that the experimental hypothesis of a direct Glu167-H (chromophore) interaction needs to be reviewed, at least when considering the Franck–Condon region of the minimum energy structure defined by this HB network. As matter of fact, it is possible that psamFP488, during its dynamics at finite temperature, can visit a configurational space compatible with a stable neutral form and a direct ESPT between Glu167 and the chromophore. However, in the minimum energy area (the Franck–Condon region) the HB network considered thus far cannot be representative of the species initiating the psamFP488 photocycle. Taken together, these results suggest that other hypotheses about the psamFP488 active site could potentially be taken into account in attempting to simulate the ESPT mechanism.

#### 2.1.2. The Chromophore in Its Neutral Form; Part II: An Alternative Model

The obtained results drove us toward the exploration of possible alternative networks and the hypothesis of a new alternative model able to represent the first species taking part to the photocycle. In Figure 8, we show our model for the protein active site. We call this model Network1.

A different scenario was considered for the reactant as well as for a possible reaction mechanism. The key difference here is that the phenolic ring O-H group does not point to the Glu167 oxygen, and is instead involved in a hydrogen bond interaction with Ser148. Ser148 interacts with Glu167, while, it is retained in one of the networks discussed before due to the water molecule arrangement. The completely different network obtained in this way could be considered as an alternative. In this way, we attempted to find a representative neutral chromophore form as well as to suggest a new hypothesis for the ESPT reaction based on indirect proton transfer between the chromophore and Glu167 modulated by Ser148.

The optimization of this new initial guess resulted in a protein in the neutral form. Although this network does not involve direct interaction between the proton donor and acceptor, and seems to resemble the GFP HB network, the important difference between psamFP488 and GFP relies on the actual spatial arrangement of the neighbor residues, which in the first case occupy a more uniform region in the space with respect to the residues around GFP, which are aligned here in a more planar arrangement. In Table 3 we show the main structural parameters obtained from the optimized structure.

The hypothesized hydrogen bond network is retained, as can be seen from the intermolecular O-O and O-H distances. A strong interaction between Ser148 and Glu167 is observed, with a O-H intermolecular distance of 1.35 Å. On the other hand, we did not observe a shortening of the distance between Glu167 and the chromophore. Indeed, an intermolecular oxygen distance of 3.19 Å is found, witth a strong interaction between Ser148 and the O-H chromophore group calculated (1.50 Å). The hydrogen bond angles confirm these results. It is interesting to observe the longer phenolic C-O distance compared to the previous cases, which is compatible with a protonated phenolic ring. Moreover, the strong hydrogen bond network found here shows oxygen intermolecular distances shorter than the ones in the previous networks, as can be observed for O(ser)-O1(glu) and O(tyr)-O(ser). However, despite this strong interaction, no proton transfer takes place.

In addition, we hypothesized another scenario in which we changed the tyrosine–water and tyrosine–Ser148 interaction, as shown in Figure 9. We call this arrangement Network2.

In this model, we further changed the chromophore’s first solvation shell. In this case, we involved its hydroxyl group in a hydrogen bond with the surrounding water molecule as a donor. Moreover, Ser148 does not directly interact with the chromophore here, instead interacting with Glu167. In this way, we avoid both the possibility of the chromophore directly interacting with Glu167 and the possibility of indirect proton transfer through Ser148, as the hydrogen bond interaction with it is ruled out. We were able to find a stationary point retaining the neutral chromophore form for this network. The main structural parameters are reported in Table 4.

The C-O distance value is again in agreement with a protonated residue form. The chromophore does not approach Glu167, and a strong hydrogen bond is observed between Ser148 and Glu167; however, in this case it involves the Glu167 oxygen labeled as O2, while the one closer to the chromophore remains O1.

This computational experiment confirms that the neutral form can be stabilized without direct interaction between the chromophore and the final proton acceptor. As expected, the residues surrounding the chromophore strongly influence its protonation state. Moreover, our results suggest a revision of the hypothesis of a direct Glu167–chromophore ESPT. Our results show that this argument may be not enough to define a protein active site network, meaning that other possible scenarios should be considered.

In Network2, the psamFP488 structure with a possible PT mechanism should involve out-of-plane motion of the phenol O-H group being suddenly activated by the excitation, leading to direct proton transfer to the Glu167. In Network2, we did not observe a satisfying saturation of the water solvation sites, which were involved in only one HB with a backbone C=O group (data not shown). Moreover, the possible PT reaction scheme involving Network2 would result in a phenolate anion product that is not stabilized by either the water molecule or Ser148, unless a further rearrangement after the excitation were considered.

On the basis of these considerations, we decided to focus on the Network1 hypothesis for further study of the ESPT reaction mechanism.

#### 2.1.3. Optical Properties of the Neutral CFP

In this section, the optical behavior of the CFP protein in its neutral form is investigated while analyzing results for both Network1 and Network2.

Table 5 shows the optical absorption values computed for the two networks. Single-point calculations were carried on the optimized structures out at the TD-B3LYP/6-31+ G(d,p)/AMBER level of theory.

Good agreement with the experimental absorption value was obtained in both cases. The experimental value of the absorption band maximum is around 407 nm.

Figure 10, Figure 11, Figure 12 and Figure 13 show the main orbitals involved in S${}_{0}$–S${}_{1}$ electronic excitation.

No important rearrangement of the chromophore electron density after the vertical excitation was found. Moreover, the main electronic reorganization substantially involves the chromophore bridge. This result is very similar to that found for the GFP [11], with no particular influence of the network arrangement on the electronic transition observed. On the basis of the orbital analysis, we can corroborate the fact that, in principle, both the networks are plausible for a neutral form of the protein, and the results are not sufficient to discriminate among them.

#### 2.1.4. Chromophore Anionic Form Characterization

In this section, we discuss the other species involved in the photocycle. More specifically, the chromophore anionic form in both the ground and the S${}_{1}$ excited state are analysed. Starting from the optimized neutral protein in Network1, we build the corresponding anionic form based on a two-proton-transfer reaction, then both the ground and first singlet electronic states of the whole protein system are optimized. In this way, we obtain the minimum on the S${}_{1}$ Potential Energy Surface (PES), i.e. the anionic form responsible for the cyan fluorescence, and the minimum on the S${}_{0}$ PES i.e., a representative structure for anionic psamFP488 in the ground state.

Here, recall that in order to investigate the proton back-transfer reaction of the photocycle, we need to explore the anionic form in its ground state.

In Figure 14 and Figure 15, we show the active site of the anionic chromophore optimized in both the electronic states, while in Table 6 we report the main structural parameters for these species.

From an inspection of the values reported in this table, it is clear that there are no important structural differences between the anions. In particular, the C-O distance is in agreement with an anionic species; moreover, the anion is stabilized by a strong interaction with Ser148 and the water molecule, showing intermolecular O-H distances of 1.73 and 1.80 Å, respectively. In particular, the last is shorter than the one found in the ground state, confirming a stabilizing role for the anionic form upon the excitation. As we show in the next section, this is further corroborated by the molecular dynamics of the ESPT reaction. Moreover, Glu167 is not as close to the phenolic ring oxygen and Ser148 as compared to the neutral form.

#### 2.1.5. ESPT Reaction Mechanism Investigation

In this section, we discuss our investigation of the ESPT reaction mechanism employing excited state ab initio molecular dynamics simulations. As underlined before, we focus on the protein model in which the proposed Network1 is adopted to describe the chromophore and arrangement of the surrounding residues.

We start with a structure representative of the minimum found in the ground state of the potential energy surface, as this provides an active site substantially ready for the reaction thanks to the structural interactions among the residues already being optimized. Here, we are able to make a number of interesting observations. In the following, we discuss the evolution in time of sevral important structural parameters involved in the ESPT reaction coordinates. Figure 16 shows the intermolecular O${}_{donor}$-H distances of the two donor–acceptor pairs, namely, the chromophore–Ser148 and Ser148–Glu167 couples. Figure 17 shows the corresponding O${}_{acceptor}$–H distance and Hwat–Otyr distance.

We observed the ESPT reaction taking place in about 30 fs. This is reasonable, as we started from a network showing strong hydrogen bond interactions. The first important result clearly observed is that the reaction mechanism is concerted; indeed, the protons substantially move at the same time. By inspection of the water–chromophore interaction, it is clear that the water molecule approaches the deprotonated phenolic oxygen after the reaction event at about 125 fs, confirming the important role of the water molecule and making less plausible an arrangement such as the one found for Network2 in which the water molecule is not able to stabilize the anion.

Figure 18 shows the evolution in time of the chromophore C-O distance and chromophore N-C-C-C dihedral angle.

Here, a shortening of the C-O group can be observed, suggesting the formation of the anionic form in agreement with the results obtained from the optimization calculations. Concerning the N-C-C-C dihedral angle, it is interesting to observe that there is a distortion from the planarity starting when the reaction takes place that is retained when the anion is formed. It is likely that this out-of-plane motion favors the reaction by placing the chromophore in the right configuration to interact with the acceptor residue; it is probably favored by the anionic chromophore. Although in this case the reaction is ultrafast, it is nonetheless possible to observe this distortion.

The ultrafast nature of this reaction (ESPT time ∼30 fs) is compatible with the reaction observed in the Franck–Condon region, with the HB interactions being prompted to support the reaction when the electronic density rearrangement of the photoacidic chromophore takes place. The experimental reaction time is necessarily larger due to the average reaction times corresponding to several structures originated by the dynamics at finite temperature. The 170 fs lifetime does correspond to a frequency compatible with collective HB stretching motions, such as those activated in the double (indirect) proton transfer proposed here. Recall, however, that this time is considerably shorter than that observed in GFP, where slower and collective frequency modes are involved, indicating a more complex mechanism. However, the proposed mechanism, although compatible with important experimental evidence, does not exclude other possibilities such as direct transfer. Using a theoretical computational approach, it is possible to investigate the nature of this mechanism and to obtain atomistic detail of the event regarding the single species involved in the reaction.

## 3. Materials and Methods

### 3.1. psammFP488 Model Preparation

The coordinates of the yellow fluorescent protein zFP538 (PDB code: 1XA9) [17] isolated from the button polyp *Zoanthus* sp. were augmented with hydrogen atoms using the program MolProbity from Richardson group [22]. All the residues were protonated according to a neutral pH, paying further attention to the residues surrounding the chromophore, such as His202. This choice is in line with the modeling of FPs, in particular of GFP, which shows the same psam chromophore and features several analogies with its binding pocket. Met167 was mutated in Glu167 according to the occupancy of this position in psamFP488. The choice of Glu167 in its deprotonated form was motivated by the close proximity of the chromophore, with this latter hypothesized to be in the neutral protonated form according to the maximum absorption value of psamFP488. Concerning the crystallographic water molecules, all the external ones were discarded, while among the internal water molecules only the one hydrogen-bonded to the chromophore was retained (the water oxygen is represented as the red sphere in Figure 3).

The choice of the zFP538 crystallographic structure for building our protein model is corroborated by the results reported in literature. In particular, it has been shown that the sequence identity between psamFP488 and zFP538 is 61%; thus, it seemed reasonable to start from this crystallographic structure when building our model.

After obtaining the protein model, we performed a partitioning of the whole system according to the ONIOM scheme [23,24,25,26]. On the basis of a multilayer approach, we employed a QM description of the chromophore (including the residue Gln66) and of the residues directly involved in the reaction or able to perform hydrogen bonds with the chromophore (a crystallographic water molecule, Ser148, and Glu167), while the rest of the system was treated at molecular mechanics (MM) level of theory employing the AMBER force field [27]. An analogue partition in GFP has been proven to reproduce the structure and relative energies between protonated and deprotonated forms of larger partitions in both the ground and excited states. Moreover, such partitioning allows for feasible simulation of the system in the excited state at TDDFT level [11,12,13]. AMBER partial charges were employed for all the residues treated at the level of MM theory, while for the chromophore we employed the parameters specifically developed by Reuter [28]. After checking the protein neutrality, it was not necessary to add any counterions. Concerning the multilayer partitioning, ad hoc cuts at the interface between the QM/MM layers were performed on single bonds of nonpolar or slightly polar bonds; in the most of these cases, C-C bonds were involved. The electronic embedding [24,29] scheme was employed to account for interactions between the two layers.

### 3.2. ESPT Network Optimization and Calculation of Optical Properties

Geometry optimization of the built system were performed at both the time-independent (DFT) and time-dependent (TD-DFT) theory levels. In particular, the ONIOM B3LYP [30] /6-31+g(d,p)/Amber theory level was employed. All the optimization calculations were performed in gas phase. This choice was made because, as a first approximation, it is reasonable that the solvent does not significantly affect the active site properties, both structural and optical. Moreover, concerning the ab initio molecular dynamics (AIMD) simulations, adding the implicit solvation to an already huge system could make our simulations very time consuming, and in the first instance it is not crucial to describe the ESPT reaction.

In particular, we optimized both the neutral and anionic CFP models, the last one in both the ground (S${}_{0}$) and first singlet (S${}_{1}$) excited state. After obtaining the neutral CFP optimized structures in the ground state, we reproduced the absorption values of this species by employing the linear response TD-DFT theory level.

### 3.3. Ab Initio Molecular Dynamics Simulations

The mechanism of the ESPT reaction was investigated by employing an excited state ab initio molecular dynamics approach. In particular, we simulated the direct proton transfer in the electronic excited state. A a neutral CFP configuration extracted from the previous ground state optimization calculations was employed as the starting configuration. Initial velocities were assigned randomly according to an average temperature of 298 K.

The excited state energies and gradients were calculated on the fly using the linear response TD-DFT [31,32,33,34] formalism at M052X/6-31+G(d,p) level of theory for the simulation. These trajectories were collected for less then 1 ps with a time step of 0.5 fs. The simulations were long enough to observe the ESPT reaction and make reasonable hypotheses with respect to the reaction mechanism.

## 4. Conclusions

In the present work, we studied the photo-induced behavior of a new cyan fluorescent protein called psamFP488. In particular, we adopted a theoretical-computational approach in order to investigate possible alternative photocycle networks, starting from experimental hypotheses and then simulating the ESPT. We were not able to find a representative network with a neutral chromophore in the ground state when adopting the experimentally hypothesized arrangement implying a direct ESPT between the choromophore and Glu167. This evidence suggested that the experimental hypothesis of a direct Glu167–H(chromophore) interaction probably needed to be reviewed, and that a scenario involving a different hydrogen bonds network had to be considered.

Therefore, we investigated possible alternative models to represent the first species taking part in the photocycle with a hydrogen bond pattern around the chromophore different from that experimentally hypothesized. The chosen arrangement (Network1) was able to properly reproduce the optical properties of the chromophore in its neutral form, corroborating the choice of this form as a possible starting arrangement for the ESPT reaction. On the basis of this newly proposed neutral form of the protein, it could be possible that the ESPT may not happen through direct interaction between donor and acceptor. Indeed, an excited state reaction mechanism involving a PT from the chromophore to Glu167 could be modulated by the Ser148 acting as a proton transfer relay station. Our study suggests that the experimental hypothesis of a direct Glu167–H(chromophore) interaction is less plausible considering the Franck–Condon region of the minimum energy structure defined by this HB network, although it is possible that, considering its dynamics at finite temperature, psamFP488 can visit a configurational space compatible with a stable neutral form and a direct ESPT between Glu167 and the chromophore.

Finally, the ESPT reaction mechanism was investigated by employing ab initio molecular dynamics. Our simulations suggested that the ESPT reaction could take place in a concerted way in a few tens of fs, with two protons moving from the chromophore to Ser148 and from Ser148 to Glu167 at the same time. These results suggest that the proposed hydrogen bond network in the ESPT could be plausible, and that the ultrafast ESPT could take place through indirect proton transfer within a tight hydrogen bond network.

## Figures and Tables

**Figure 1 ijms-23-15640-f001:**
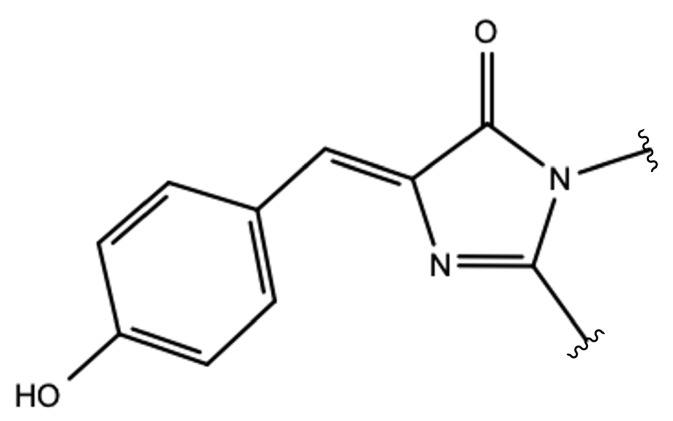
Schematic picture of the neutral HBDI chromophore.

**Figure 2 ijms-23-15640-f002:**
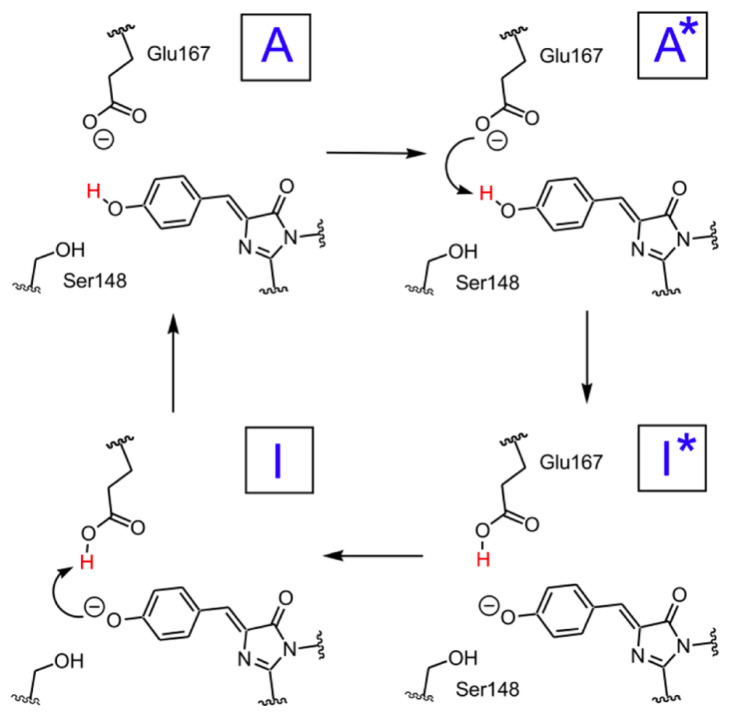
Schematic representation of the hypothesized psamFP488 photocycle, adapted from [19]. Chromophore neutral forms in ground and excited state are labeled as A and A${}^{*}$, respectively, while chromophore anionic forms are respectively labeled as I and I${}^{*}$.

**Figure 3 ijms-23-15640-f003:**
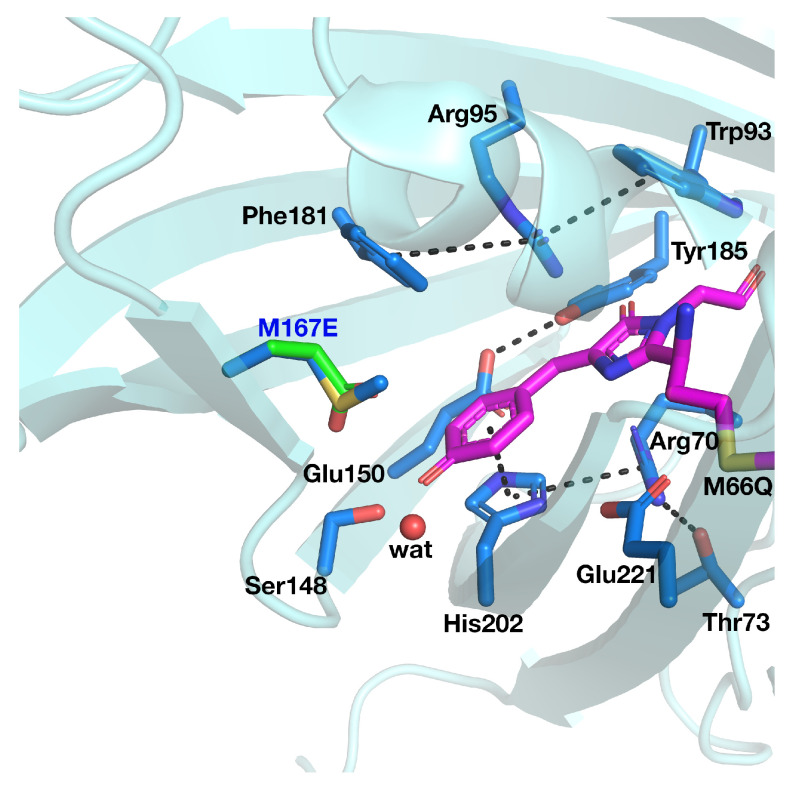
Active site of the psamFP488 homologous zFP538-K66M, with the M167E mutation highlighted. In psamFP488, Met66 is also mutated (Gln66). The chromophore and surrounding residues are shown in sticks representation, and the protein is shown as a cartoon representation. Hydrogen bonds, salt bridges, and π-stacking interactions involving residues surrounding the chromophore are highlighted as black dashed lines. All hydrogen atoms are hidden for clarity.

**Figure 4 ijms-23-15640-f004:**
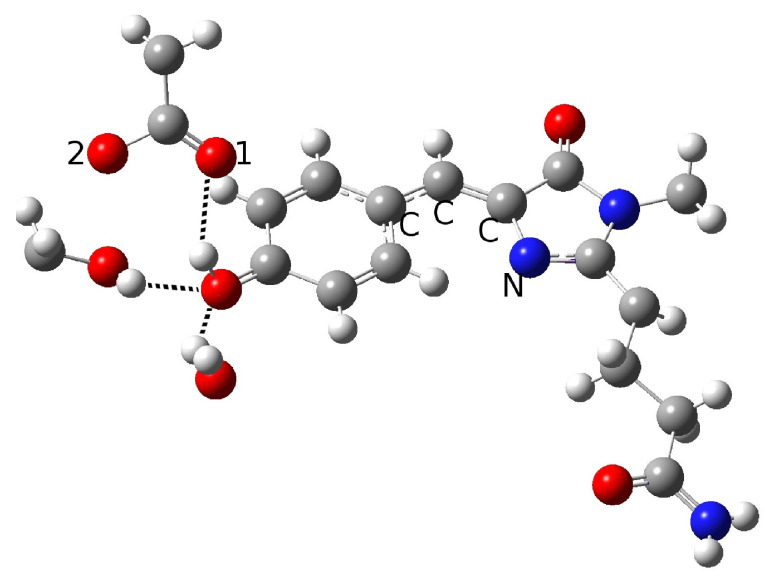
First hypothesized hydrogen bond network in neutral psamFP488 leading to ESPT on the basis of the experimental hypothesis.

**Figure 5 ijms-23-15640-f005:**
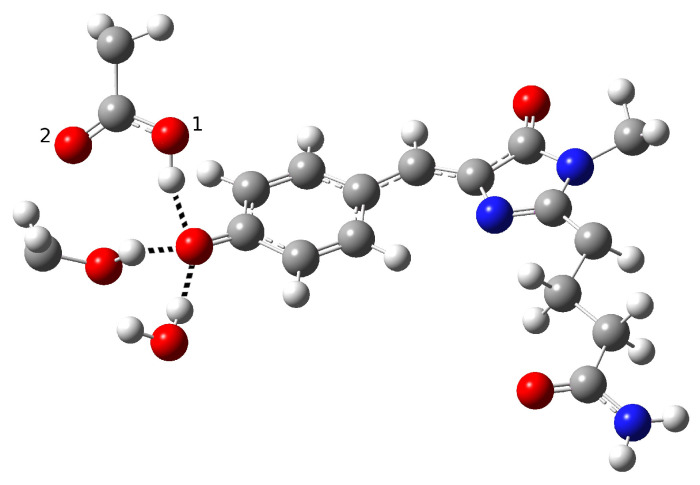
Anionic form resulting from the optimization of psamFP488 starting from the first HB network.

**Figure 6 ijms-23-15640-f006:**
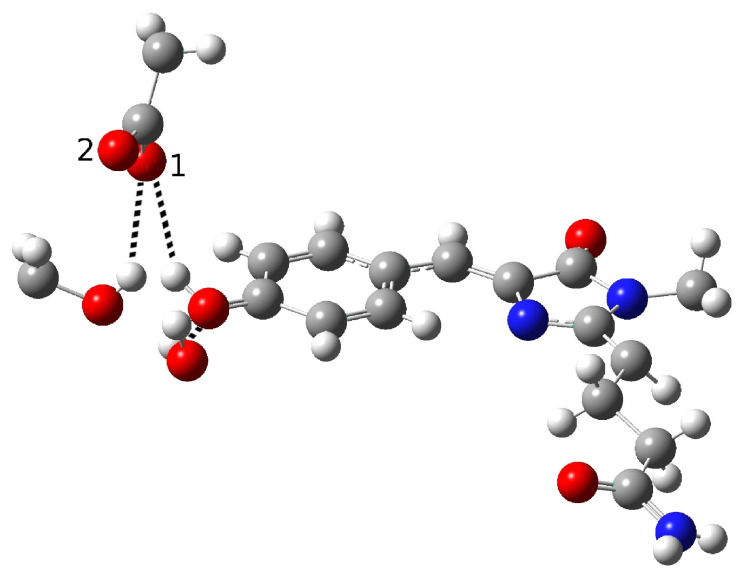
Second hydrogen bond network in psamFP488, proposed as the neutral form leading to ESPT on the basis of the experimental hypothesis.

**Figure 7 ijms-23-15640-f007:**
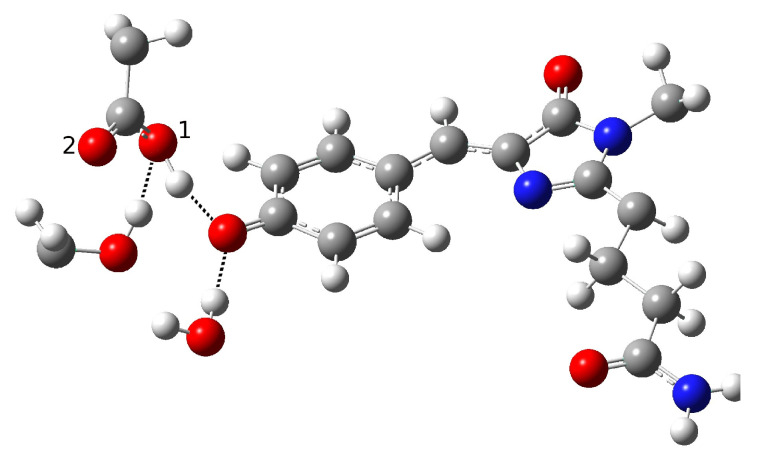
Anionic form resulting from the optimization of psamFP488 starting from the second HB network.

**Figure 8 ijms-23-15640-f008:**
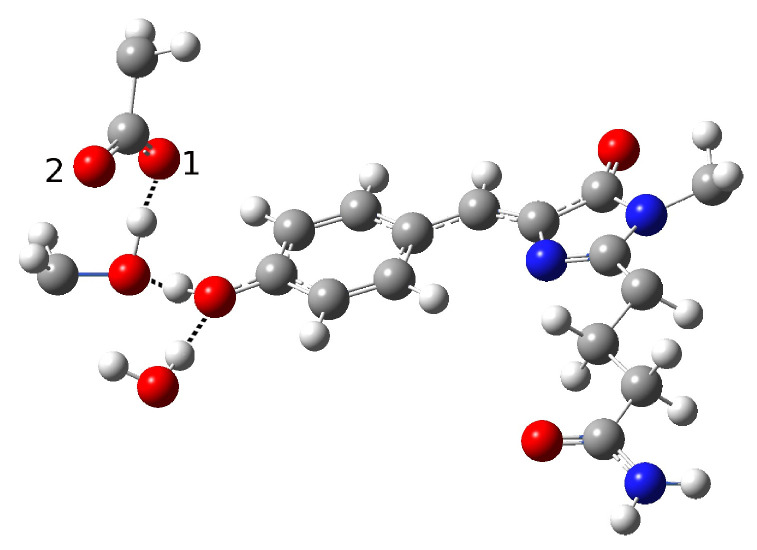
The first hydrogen bond network in psamFP488 (Network1), proposed here as the neutral form leading to ESPT.

**Figure 9 ijms-23-15640-f009:**
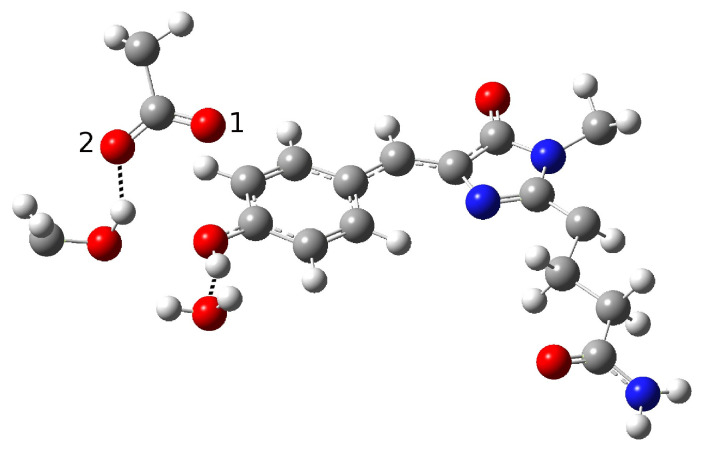
Second hydrogen bond network in psamFP488 (Network2) proposed as the neutral form leading to ESPT.

**Figure 10 ijms-23-15640-f010:**
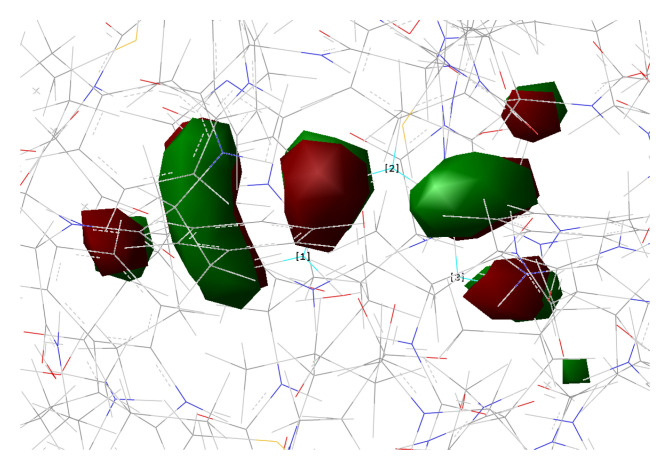
HOMO calculated for psamFP488 in Network1.

**Figure 11 ijms-23-15640-f011:**
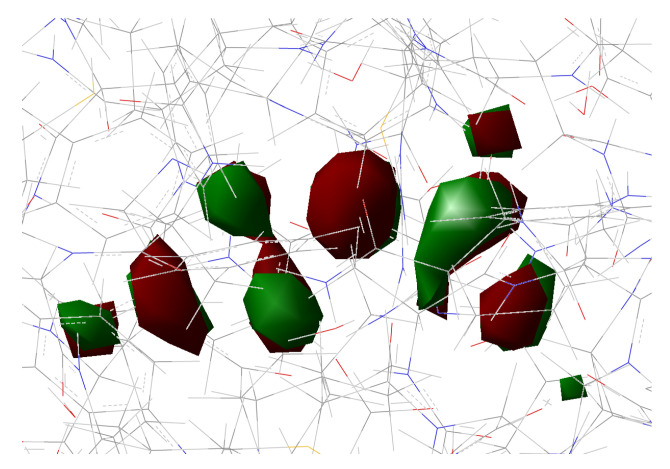
LUMO calculated for psamFP488 in Network1.

**Figure 12 ijms-23-15640-f012:**
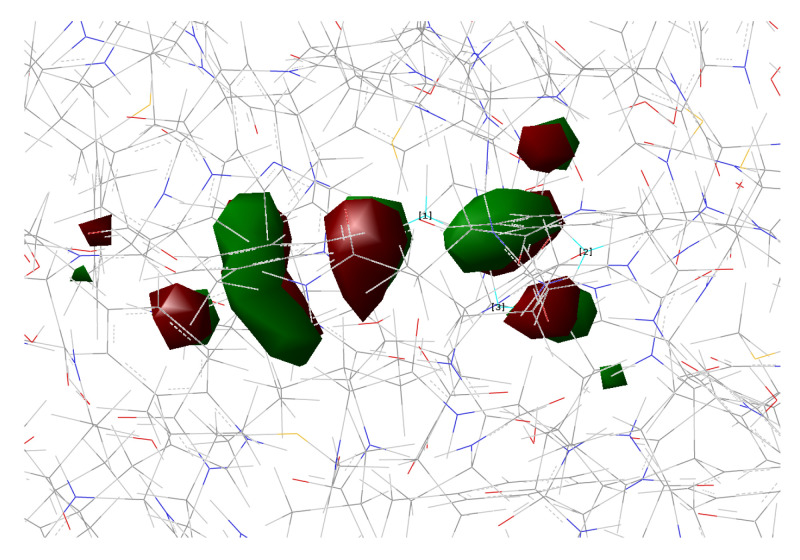
HOMO calculated for psamFP488 in Network2.

**Figure 13 ijms-23-15640-f013:**
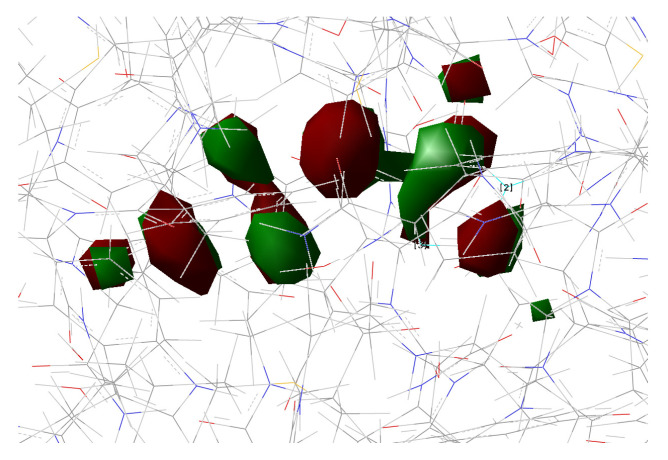
LUMO calculated for psamFP488 in Network2.

**Figure 14 ijms-23-15640-f014:**
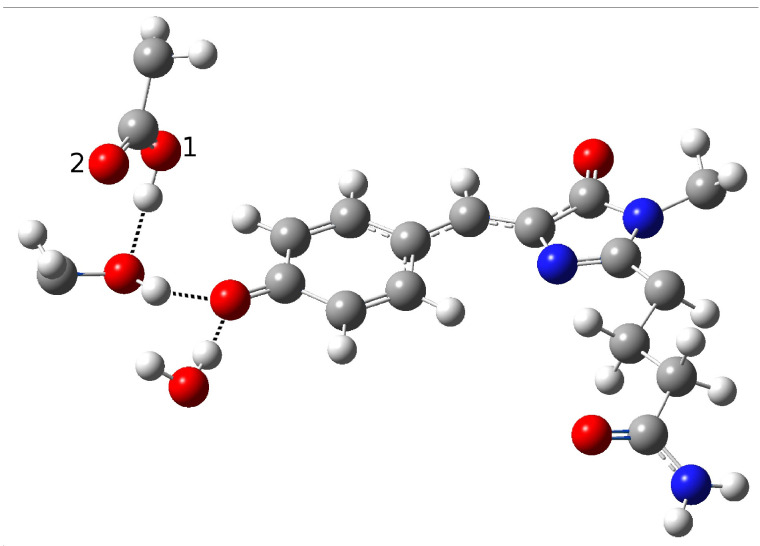
psamFP488 in anionic form optimized in the ground state.

**Figure 15 ijms-23-15640-f015:**
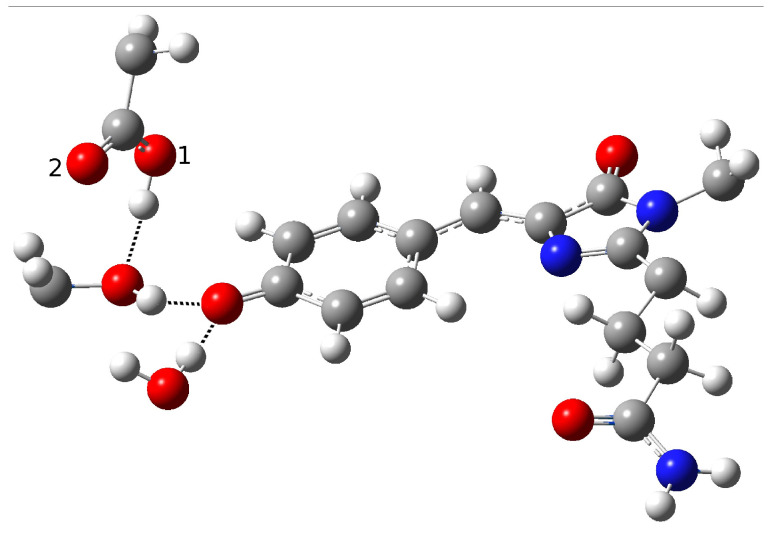
psamFP488 in anionic form optimized in the S${}_{1}$ excited state.

**Figure 16 ijms-23-15640-f016:**
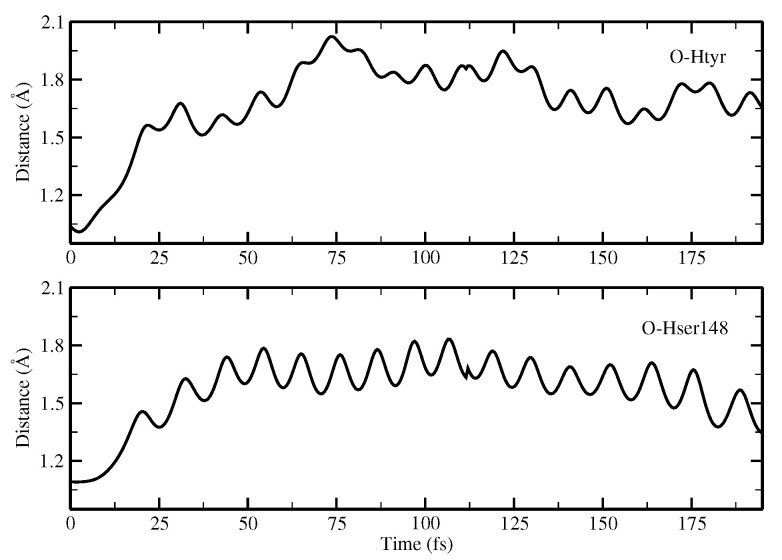
Time evolution of the donor OH distances (Å) involved in concerted PT. Upper panel: evolution of the OH(tyr) distance. Lower panel: evolution of the OH(ser) distance.

**Figure 17 ijms-23-15640-f017:**
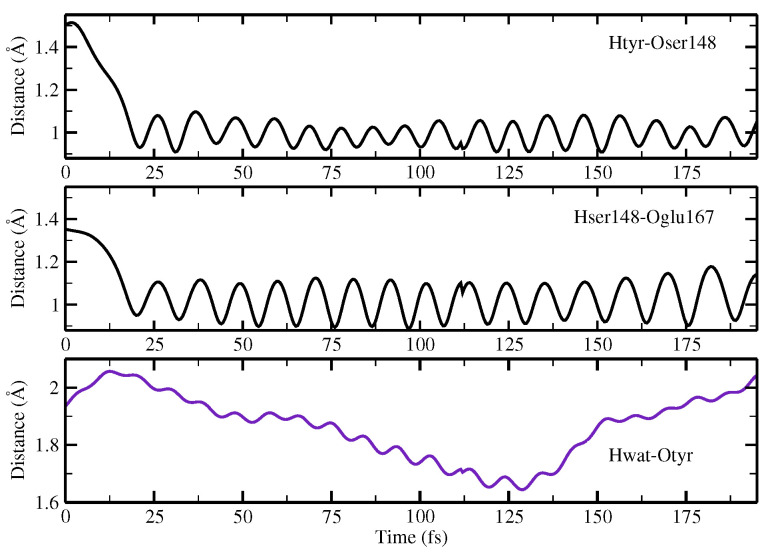
Network distance (Å) evolution involved in concerted PT showing the time evolution of the H(tyr)-O(ser) distance (upper panel), H(ser)-O(glu) distance (middle panel), and H(wat)-O(tyr) distance (lower panel).

**Figure 18 ijms-23-15640-f018:**
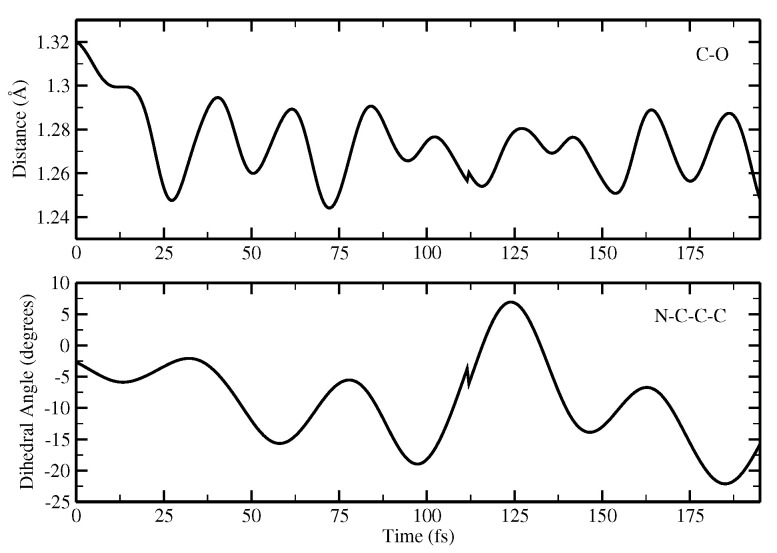
Time evolution of the CO(tyr) distance and N-C-C-C dihedral angle of the chromophore.

**Table 1 ijms-23-15640-t001:** Structural parameters (distances (Å) and angles (degrees)) of psamFP488 optimized starting from the first HB network.

	Distance (Å)
O(water)-O(tyr)	2.894
O(tyr)-O(ser)	2.794
O(water)-O(ser)	4.390
O(tyr)-O1(glu)	2.661
O(tyr)-O2(glu)	3.500
O(ser)-O1(glu)	4.282
O(ser)-O2(glu)	3.509
C−O(tyr)	1.282
O-H(glu)	0.994
H(water)-O(tyr)	1.917
H(ser)-O(tyr)	1.842
	Angle (degrees)
H-O(water)-O(tyr)	2.73
H-O(ser)-O(tyr)	10.25
H-O(glu)-O(tyr)	13.16

**Table 2 ijms-23-15640-t002:** Structural parameters (distances (Å) and angles (degrees)) of psamFP488 optimized starting from the second HB network compatible with the experimental hypothesis.

	Distance (Å)
O(water)-O(tyr)	2.909
O(tyr)-O(ser)	3.096
O(water)-O(ser)	4.617
O(tyr)-O1(glu)	2.479
O(tyr)-O2(glu)	3.602
O(ser)-O1(glu)	2.944
O(ser)-O2(glu)	4.072
C-O(tyr)	1.277
O-H(glu)	1.036
H(water)-O(tyr)	1.933
H(ser)-O1(glu)	2.082
	Angle (degrees)
H-O(water)-O(tyr)	2.19
H-O(ser)-O(glu)	22.32
H-O(glu)-O(tyr)	10.69

**Table 3 ijms-23-15640-t003:** Structural parameters (distance (Å) and angle (degrees)) of psamFP488 in the neutral form optimized in the Network1 arrangement.

	Distance (Å)
O(water)-O(tyr)	2.901
O(tyr)-O(ser)	2.538
O(water)-O(ser)	4.464
O(tyr)-O1(glu)	3.196
O(tyr)-O2(glu)	3.567
O(ser)-O1(glu)	2.442
O(ser)-O2(glu)	3.326
C-O(tyr)	1.320
H(water)-O(tyr)	1.934
H(tyr)-O(ser)	1.500
H(ser)-O1(glu)	1.352
	Angle (degrees)
H-O(water)-O(tyr)	4.81
H-O(tyr)-O(ser)	2.91
H-O(ser)-O(glu)	2.77

**Table 4 ijms-23-15640-t004:** Structural parameters (distance (Å) and angle (degrees)) of psamFP488 in the neutral form optimized in the Network2 arrangement.

	Distance (Å)
O(water)-O(tyr)	2.810
O(tyr)-O(ser)	2.728
O(water)-O(ser)	4.452
O(tyr)-O1(glu)	2.926
O(tyr)-O2(glu)	3.173
O(ser)-O1(glu)	4.250
O(ser)-O2(glu)	2.765
C-O(tyr)	1.311
H(tyr)-O(water)	1.853
H(ser)-O2(glu)	1.866
	Angle (degrees)
H-O(tyr)-O(water)	10.89
H-O(ser)-O(glu)	19.91

**Table 5 ijms-23-15640-t005:** Absorption values (eV and nm) calculated for psamFP488 arranged according to Network1 and Network2.

	eV	nm
Network1	3.13	396.59
Network2	3.11	398.49

**Table 6 ijms-23-15640-t006:** Structural parameters (distance (Å) and angle (degrees) involved in the anionic chromophore form of the protein optimized in ground and excited states.

	Ground State	Excited State
	Distance (Å)	
O(water)-O(tyr)	2.778	2.774
O(tyr)-O(ser)	2.679	2.680
O(water)-O(ser)	4.352	4.347
O(tyr)-O1(glu)	3.511	3.486
O(tyr)-O2(glu)	3.834	3.835
O(ser)-O1(glu)	2.648	2.653
O(ser)-O2(glu)	3.301	3.321
C-O(tyr)	1.269	1.280
H(wat)-O(tyr)	1.806	1.800
H(ser)-O(tyr)	1.734	1.736
H(glu)-O(ser)	1.648	1.652
	Angle (degrees)	
H-O(wat)-O(tyr)	5.98	6.15
H-O(ser)-O(tyr)	13.87	14.17
H-O(glu)-O(ser)	4.20	1.17

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
