# Peer review of "Direct or Indirect ESPT Mechanism in CFP psamFP488? A Theoretical-Computational Investigation"

_ijms, 2022, doi:10.3390/ijms232415640_

Round 1

Reviewer 1 Report

The paper by G. Donati et al. reports a computational study on the excited-state proton transfer in cyan fluorescent protein psamFP488, which is experimentally known to undergo photoinduced deprotonation on an ultrafast timescale of 170 fs. Here, the main challenge was to reproduce the initial ground-state configuration of the chromophore and its nearest surrounding inside the protein to ensure that the chromophore is in the neutral (protonated) state before excitation and becomes deprotonated and negatively charged in the excited state. Using standard computational protocols for QM/MM simulations and ab initio molecular dynamics and DFT/TDDFT theory for ground-state/excited-state calculations, the authors conclude that the most plausible arrangement of the hydrogen-bonded network around the chromophore includes Ser148, which serves as an intermediate in the excited-state proton transfer between the chromophore and proton acceptor Glu167. As such, the authors show that ultrafast excited-state proton transfer can occur through an indirect mechanism in fluorescent proteins, where a proton acceptor and a proton donor are not directly bound to each other. The paper is well written, and it is easy to follow. I think, the paper can become publishable in Int. J. Mol. Sci.; however, certain issues must be resolved prior to its publication.

The main question here is how well the model used to study excited-state proton transfer in psamFP488 is chosen. The QM part used by the authors represents a minimal model to enable a proton transfer between the chromophore and Glu167. At the same time, there are quite a lot of charged residues inside the chromophore’s binding pocket (shown in Fig. 3), which can affect the pKa of the chromophore and Glu167. Moreover, there are also residues such as His202 that are close to the chromophore, and it is not straightforward to assign their charged state. My question is how the authors justify their model. Is there any experimental evidence for a particular choice of the charged state of the amino acid residues close to the chromophore? Are there any results, except for the vertical excitation energy, that can directly be comparted to the experimental data? Finally, how will the size of the QM part affect the charged states of the chromophore and Glu167? The latter is of particular interest, since the first model, shown in Fig. 5, provides no stabilization of the anionic state of Glu167.

It would also be helpful to provide minimum energy pathways for proton transfer in the ground and excited states. At least, an energy diagram showing the stationary points.

Minor issues

1. The numeration of sections becomes incorrect in Materials and Methods.

2. Figures 10,11,12,13 should be redrawn, since it is difficult to understand what the authors want to show. Also, the oscillator strengths for the S0-S1 transition should be mentioned in the text.

Reviewer 2 Report

The manuscript presents a computational study that has been performed to help understanding the mechanism of ultrafast excited-state proton transfer (ESPT) in a cyan fluorescent protein (CFP) called psamFP488. In contrast to other CFP variants, psamFP488 stabilizes the protonated form of the chromophore in the ground state, undergoing an ultrafast ESPT reaction, in 170 fs, upon photoexcitation. In the ground state, back proton transfer from the protein to the chromophore takes only 110 fs. Mutagenesis studies in combination with the protein structure of a homologous CFP (the protein crystal structure of psamFP488 is not available) allowed hypothesizing that the ESPT is due to the presence of deprotonated Glu167 close to the protonated chromophore. The computational study described in the manuscript addressed two possible proton transfer reactions; one considers a direct hydrogen bond between the chromophore and Glu167, and the other – a hydrogen bond mediated by a serine side chain. The first reaction is ruled out as the direct hydrogen bond leads to chromophore deprotonation in the ground state. For the second reaction, it is demonstrated that the excited chromophore undergos deprotonation in less than 30 fs. In addition, a structure of the active site without any hydrogen bond between the chromophore and Glu167 was optimized, but ESPT was not considered.

Overall, the study presents a successful attempt to model ultrafast ESPT using computational chemistry, whereas the majority of this kind of simulations address a double-bond isomerization reaction. This ESPT simulation is arguably enabled by the choice of the system, i.e. a rather small QM subsystem featuring close proximity of the proton donor and acceptor that ensures a short timescale of proton transfer. Clearly, the characterized ESPT manifests well-established photo-acidity of the GFP chromophore. The major question to be addressed, in my opinion, concerns the interactions with the protein (and water) stabilizing the protonated chromophore in the ground state. In GFP, this is achieved  by a distance separation of the proton donor and acceptor; in psamFP488, the distance separation is presumable much smaller than in GFP. Unfortunately, the manuscript does not attempt addressing this critical point. A proper homology modelling of the psamFP488 structure, which would have been necessary to answer the question, has not been attempted. Even within the chosen restricted approach, the authors could have compared the stability of the protonated chromophore in Network 1. Such a study is straightforward – ground-state dynamics simulations starting from the deprotonated chromophore structure must reveal (to agree with the experiment) an ultrafast re-protonation of the chromophore. Unfortunately, such simulation is missing and without it, one cannot conclude that Network 1 represents the likely arrangement in the ground state. Moreover, in my opinion, Network 1 is inconsistent with the experimentally observed ESPT dynamics. The computations indicate (Figures 17 and 18) that deprotonation occurs in a single vibration period (⁓25 fs corresponds to ⁓1300 cm-1), whereas the experimentally determined 170 fs lifetime indicates a qualitatively different process. Given these issues, the conclusions are supported by the data only partially, and in, my opinion, the ESPT reaction in psamFP488 could not been identified yet. I suggest that the authors carefully explain all limitations of their modeling and more carefully reformulate the conclusions.

Round 2

Reviewer 1 Report

I am satisfied with the changes made by the authors to the revised version of the manuscript. The paper needs to be corrected for typos, but otherwise I can recommend it for publication in IJMS.

Reviewer 2 Report

The authors have satisfactorily dealt with my comments on the original version of their manuscript.